
# Optimal control for Hamiltonian parameter estimation in non-commuting and bipartite quantum dynamics

Shushen Qin[1,2*], Marcus Cramer[3], Christiane P. Koch[4] and Alessio Serafini[1]

**1** Department of Physics & Astronomy, University College London,
Gower Street, London WC1E 6BT, UK
**2** Centre for Quantum Technologies, National University of Singapore, Singapore
**3** Q-CTRL, Sydney, NSW Australia & Berlin, Germany
**4** Fachbereich Physik and Dahlem Center for Complex Quantum Systems,
Freie Universität Berlin, Arnimallee 14, 14195 Berlin, Germany

⋆ shushen.qin@u.nus.edu

## Abstract

The ability to characterise a Hamiltonian with high precision is crucial for the implementation of quantum technologies. In addition to the well-developed approaches utilising optimal probe states and optimal measurements, the method of optimal control can be used to identify time-dependent pulses applied to the system to achieve higher precision in the estimation of Hamiltonian parameters, especially in the presence of noise. Here, we extend optimally controlled estimation schemes for single qubits to non-commuting dynamics as well as two interacting qubits, demonstrating improvements in terms of maximal precision, time-stability, as well as robustness over uncontrolled protocols.

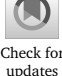

# 1   Introduction

For a quantum device to function accurately, according to the purpose of its design, the parameters characterising the system dynamics must be known as precisely as possible. A central task of quantum metrology is to devise schemes to reach the ultimate precision achievable in the estimation of a parameter for a particular system, especially in the presence of noise. First established in the late sixties [1], quantum estimation techniques have come to the fore with the advent of quantum information science [2], with the identification of the "standard" and ultimate "quantum" limits on estimation [3–6], and extensive application to noisy open quantum systems [7–12], as well as to a vast number of specific variant schemes and systems [13–17].

Typically, the estimate of an unknown parameter $x$ involves preparing a probe state $\rho_0$ and evolving it under a parameter-dependent dynamical process $\mathcal{H}_x$. Then, performing a POVM on the final state $\rho_x$ allows one to obtain an estimate of the parameter value $\hat{x}$ from the measurement outcomes [15]. The quantum Cramér-Rao bound sets the ultimate bound to the precision of such an estimate: $\delta\hat{x} \geq \frac{1}{\sqrt{F_Q(x)}}$, where $\delta\hat{x}$ is the standard deviation of an unbiased estimator of $x$ and $F_Q(x)$ is the 'quantum Fisher information' (QFI), defined as [5]:

$$F_Q(x) = Tr\left(\rho_x \ell_x^2\right),$$

where $\ell_x$ is a self-adjoint, symmetric logarithmic derivative (SLD) operator satisfying $\partial_x \rho_x = \frac{1}{2}\{\ell_x, \rho_x\}$. Maximising the Fisher information leads to minimising this lower bound, resulting in a more precise estimate.

In order to reach the ultimate precision limit set by the physical system, much efforts have been made in the development of standard metrological approaches involving the preparation of optimal probe states [4,18–23] and the execution of optimal measurements [24–29]. Using $N$ identical, independent probes, when $N$ is sufficiently large, the error in the estimated value scales as $1/\sqrt{N}$ in accordance with the central limit theorem. This defines the shot-noise limit, which is the limit on the estimation precision of a classical scheme. The use of quantum effects, such as probing with entangled or squeezed states, may further improve on this limit, to reach the more fundamental Heisenberg limit, arising from the Heisenberg uncertainty relation setting a lower bound on the simultaneous measurement of incompatible observables: If an $N$-entangled probe state is used, the ultimate precision of an estimate scales as $1/N$ [3,4]. However, it is usually difficult to prepare a large, entangled state, which makes the benefit of this quantum-enhanced parallel scheme difficult to harness in practice. Hence attention has recently turned to sequential schemes too, where a probe state evolves under the same dynamics multiple times, which reaches the same precision as the parallel scheme under unitary dynamics, and outperforms the parallel scheme for certain parameter estimation tasks [30].

It is known that under unitary dynamics, when the whole Hamiltonian takes a multiplicative form of the parameter, $\mathcal{H}(x) = x\mathcal{H}$ (as in the case of "phase estimation"), the optimal probe state is of the form $(|\lambda_{max}\rangle + |\lambda_{min}\rangle)/\sqrt{2}$, where $|\lambda_{max}\rangle$ and $|\lambda_{min}\rangle$ are eigenvectors of $\mathcal{H}$ corresponding to its maximum and minimum eigenvalues [4]. Under unitary dynamics, the optimal QFI value attainable is given by $F_Q = (\lambda_{max} - \lambda_{min})^2 T^2$, which corresponds to the Heisenberg scaling of the precision limit for a continuous dynamic at evolution time T [31]. In this case, the precision improves asymptotically with time, scaling as $1/T$. However, for more general non-unitary and non-commuting dynamics (i.e., when the Hamiltonians at two different values of the parameter to be estimated do not commute with each other), controlled sequential schemes are required to achieve a similar scaling [30–32].

Indeed, estimation schemes may in general be complemented and refined through the use of quantum controls, in the form of tunable Hamiltonian terms that allow one to alter the intermediate dynamics of a system and to achieve a higher estimation precision. Dynamical decoupling techniques have been applied to improve noisy parameter estimation [33,34].

The use of periodic unitary rotations was shown to achieve QFI values even exceeding the Heisenberg scaling, for time-dependent Hamiltonian within a short estimation duration limited by the system coherence time [35–37]. Control-enhanced parameter estimation has also been considered for continuous variable systems [38, 39].

The effectiveness of quantum controls in improving the precision limit attainable motivates inquiries into systematic strategies to identify for optimal control fields across a variety of systems. Previous work has demonstrated the effectiveness of the controls obtained by numerical optimisation [40] for the estimation of multiplicative Hamiltonian terms under non-unitary evolutions [32, 41–44]. So far, most of these studies have been limited to single-qubit systems (with the notable, very recent exception of [44], where Pontryagin's maximum principle is applied to a twist and turn Hamiltonian in arbitrary dimension). In view of the role estimation is bound to play in quantum technologies, it would be highly desirable to address the estimation of Hamiltonian parameters in the presence of non-negligible interactions between controlled quantum systems. To this aim, the present study extends previous investigations to more general, non-commuting single-qubit dynamics as well as to the estimation of both local and interaction terms in two-qubit dynamics. We will show that, in most cases, the use of Hamiltonian controls grants a definite advantage towards such tasks, in terms of both maximal precision and stability.

This paper is organised as follows. In Section 2, we define the theoretical framework. In Section 3, we apply such optimisation to single-qubit systems, in the presence of dephasing (Sec. 3.1) as well as relaxation (Sec. 3.2), and then move on to consider the non-commuting case of estimating the magnetic field's direction, rather than its magnitude. In Section 4, we consider noisy two-qubit systems, addressing both the estimation of local frequencies (Sec. 4.1) and interaction strengths (Sec. 4.2) through local measurements. We draw conclusions in Section 5.

## 2 Optimal Control via Krotov's method and Hilbert-Schmidt minimisation

The dynamics of the systems considered for this study is described by the general master equation:

$$\dot{\rho} = \mathcal{L}(\rho) = -\frac{i}{\hbar}[\mathcal{H}, \rho] + \sum_j \left( L_j \rho L_j^{\dagger} - \frac{1}{2}\{L_j^{\dagger} L_j, \rho\} \right) = -\frac{i}{\hbar}[\mathcal{H}, \rho] + \mathcal{L}_D(\rho) \,, \quad (1)$$

whose Lindblad operators will be specified as we deal with different cases. The system Hamiltonian is written as:

$$\mathcal{H} = \mathcal{H}_0 + \sum_j u_j(t) \, \mathcal{H}_j^c \,, \quad (2)$$

where the drift Hamiltonian $\mathcal{H}_0$ governs the original, uncontrolled system, while the set of control Hamiltonians $\mathcal{H}_j^c$, with time-dependent coefficients $u_j(t)$, represent the externally tunable control fields. A quantum control problem can then be formulated as a search for a set of controls $u_j(t)$ that optimises a cost functional quantifying the control objective.

A general cost functional $\mathcal{J}$ consists of a terminal cost term $\mathcal{J}_T$, which is a figure of merit to determine how close the controlled dynamics is to the control objective at the final time $T$ of a system evolution, and a running cost term $g$ which accounts for time-dependent costs at intermediate times:

$$\mathcal{J}\left(\rho, \{u_j\}\right) = \mathcal{J}_T[\rho(T)] + \int_0^T g\left[\{\rho(t)\}, \{u_j(t)\}\right] dt \,, \quad (3)$$

where $\rho$ is the forward propagating state. The task of improving the estimation precision could be set as a control problem with the control objective of maximising the associated QFI at the end of an estimation protocol with a target time $T$ [32]. To this aim, it is useful to note that the precision of a parameter's estimation is reflected in the distinguishability between the neighbouring states $\rho_x$ and $\rho_{x+\delta x}$, obtained by evolving the same probe state $\rho_0$ under Hamiltonians $\mathcal{H}_x$ and $\mathcal{H}_{x+\delta x}$ under an infinitesimally small change in the parameter of interest. The QFI can indeed be directly determined in terms of a distance measure [2]:

$$F_Q(x) = \frac{4D_{Bures}^2(\rho_x, \rho_{x+\delta x})}{dx^2}, \tag{4}$$

where $D_{Bures}(\rho_x, \rho_{x+\delta x}) = \sqrt{2 - 2F_B(\rho_x, \rho_{x+\delta x})}$ is the Bures distance between $\rho_x$ and $\rho_{x+\delta x}$, and $F_B(\rho_x, \rho_{x+\delta x}) = Tr\left(\sqrt{\sqrt{\rho_x}\rho_{x+\delta x}\sqrt{\rho_x}}\right)$ is their fidelity. Hence, maximising the QFI is equivalent to maximising the distance between the states $\rho_x(T)$ and $\rho_{x+\delta x}(T)$ at the end of evolution over a duration $T$. However, the Bures distance is highly non-linear and difficult to optimise in practice. So, following the steps of previous inquiries on this subject [42,43], we define instead a cost functional adopting the Hilbert-Schmidt (HS) distance as an alternative, simpler distance metric between the state $\rho_x$ and its neighbouring state $\rho_{x+dx}$:

$$\mathcal{J}_T = 1 - \frac{1}{2}Tr[(\rho_x - \rho_{x+\delta x})^2]. \tag{5}$$

While the minimisation of $\mathcal{J}_T$ defined using the HS distance does not guarantee of improving the QFI, there is numerical evidence that optimisation performed using the HS distance leads to a similar optimisation of other distance measures, such as the Bures distance, which relates directly to the QFI [45]. Thus, the Hilbert-Schmidt criterion provides one with an viable numerical target, whose efficacy will be assessed *a posteriori*, by evaluating the QFI directly through Eq (4).

The numerical minimisation of the Hilbert-Schmidt distance will be carried out through the gradient-based, monotonically convergent Krotov's method [46,47]. The system dynamics is discretised in time with piece-wise constant control amplitudes for which the values are updated through each iteration of the algorithm. The running cost term is assumed to take the form:

$$g\left[\{\rho(t)\}, \{u_j(t)\}\right] = \sum_j \frac{\lambda_j}{S_j(t)}\left[u_j^{(r+1)}(t) - u_j^{(r)}(t)\right]^2, \tag{6}$$

where the index $(r+1)$ indicates the current iteration, $\lambda_j$ is a numerical parameter corresponding to the step size in the update of the control and $S_j(t) \in [0,1]$ is a shape function which switches the control on and off smoothly. The form of Eq. (6) is a standard choice to prevent the pulse from drifting towards unphysical shapes. The control amplitudes are updated through iterations with the following rule:

$$\Delta u_j(t) = \frac{S_j(t)}{\lambda_j}\Re\left(\left\langle \chi^{(r)}(t), \frac{\partial \mathcal{L}(\rho, \{u_j\})}{\partial u_j}\bigg|_{u_j^{(r+1)}(t)}\rho^{(r+1)}(t)\right\rangle\right). \tag{7}$$

Here, $\chi^{(r)}(t)$ is a costate that is backward propagated for the time interval $[T, t]$ under the old control field, while $\rho^{(r+1)}(t)$ is a forward propagating state for the time interval $[0, t]$ under the new field. The iteration process terminates when $\Delta u_j(t)$ tends to zero as the forward propagated state gets closer to the target, and the control field converges to an optimised solution. The running cost $g$ then vanishes and the overall cost functional $\mathcal{J}$ only has the $\mathcal{J}_T$ term representing the optimised figure of merit. The system dependence with respect to the control is:

$$\frac{\partial \mathcal{L}(\rho)}{\partial u_j}\bigg|_{\rho^{(r+1)}, u_j^{(r+1)}} = -\frac{i}{\hbar}\left[\frac{\partial \mathcal{H}}{\partial u_j}\bigg|_{u_j^{(r+1)}}, \rho^{(r+1)}\right] = -\frac{i}{\hbar}\left[\mathcal{H}_j, \rho^{(r+1)}\right]. \tag{8}$$

Since $\mathcal{H}$ has linear coupling to the control, the explicit dependence on $u^{(r+1)}$ vanishes. There is only implicit dependence via $\rho^{(r+1)}$, which may be simplified by lowest order approximation and the right choice of time discretisation. The forward propagating state $\rho(t)$ under the influence of the updated control pulse has the following dynamic:

$$\frac{\partial \rho^{(r)}}{\partial t} = -\frac{i}{\hbar}\left[\mathcal{H}\left(u^{(r)}(t)\right), \rho^{(r)}\right] + \mathcal{L}_D\left(\rho^{(r)}\right), \tag{9}$$

with initial condition: $\rho^{(r)}(0) = \rho_{initial}$. The backward propagating $\chi(t)$ under influence of the pulse from the previous iteration is solved backward in time:

$$\frac{\partial \chi^{(r)}}{\partial t} = -\frac{i}{\hbar}\left[\mathcal{H}\left(u^{(r)}(t)\right), \chi^{(r)}\right] - \mathcal{L}_D^\dagger\left(\chi^{(r)}\right), \tag{10}$$

with an 'initial' condition: $\chi^{(r)}(T) = -\nabla_{\rho(T)}\mathcal{J}_T$, derived from the target state at final time $T$. The coupled control equations are solved simultaneously for each iteration, starting with an initial guess of $u^{(0)}(t)$ [48]. All control optimisations were initiated with an arbitrary guess field of a small constant value that was switched on and off smoothly around $t = 0$ and $t = T$. This feature is preserved throughout the optimisation by the definition of the shape function $S_j(t)$. After the first run of the optimisation algorithm which yields a preliminary optimal control, prominent features from the preliminary control are fed back into the algorithm as a new guess field to ensure convergence to a smooth and regular control field.

# 3 Application to a single-qubit

Here, the effects of optimal control to improve the QFI is considered for estimating a parameter of a magnetic field acting on a single qubit. It should be noted that subsections 3.1 and 3.2, dealing with the estimation of a magnetic field amplitude along a known direction under dephasing and relaxation respectively, do not make any conceptual advance on the results of Ref. [42]; they were included to set the stage and serve as a benchmark for further inquiry. The system considered is a spin-1/2 probe placed in a magnetic field of amplitude $B$, applied along a direction defined by $\omega$, in the presence of decoherence by either dephasing or relaxation. The free, noiseless evolution of the spin-1/2 particle in the magnetic field would be governed by:

$$\mathcal{H}_0 = \frac{B}{2}(\sin(\omega)\sigma_x + \cos(\omega)\sigma_z). \tag{11}$$

The control employed is applied in all directions, and described by $\mathcal{H}_c = \vec{u}(t) \cdot \vec{\sigma}$, with $\vec{u}(t) = \left(u_x(t), u_y(t), u_z(t)\right)$ being the magnitude of the continuous control fields in each direction and $\vec{\sigma} = \left(\sigma_x, \sigma_y, \sigma_z\right)$ being the Pauli matrices.

Since the scope of this study is focused on single parameter estimation, we first set $\omega = 0$ in Eq (11). In this case, $\mathcal{H}_0 = \frac{1}{2}B\sigma_z$, aligning the magnetic field along the $z$ axis, and the parameter of interest $B$ represents the magnitude of the magnetic field, assumed to have a true value $B_0$. The optimal probe state used is $|+\rangle = (|0\rangle + |1\rangle)/\sqrt{2}$, with an associated scaling of the QFI given by $F_Q = T^2$. Intuitively, the $|+\rangle$ state lies on the $x$-$y$ plane of the Bloch sphere, which is most sensitive to a magnetic field in the $z$ direction [43].

The distinguishability of two neighbouring states with a slight difference in the value of the parameter $B$ arises from a difference in the rate of rotation about the $z$ axis, which is dependent on the magnitude of the magnetic field. The discrepancy in relative phase accumulates as the two states evolve. The minimum time required for perfect distinguishability, i.e. for being exactly opposite on the Bloch sphere, is set by a quantum speed limit, $T_{QSL} = \frac{\pi}{\delta B}$, assuming

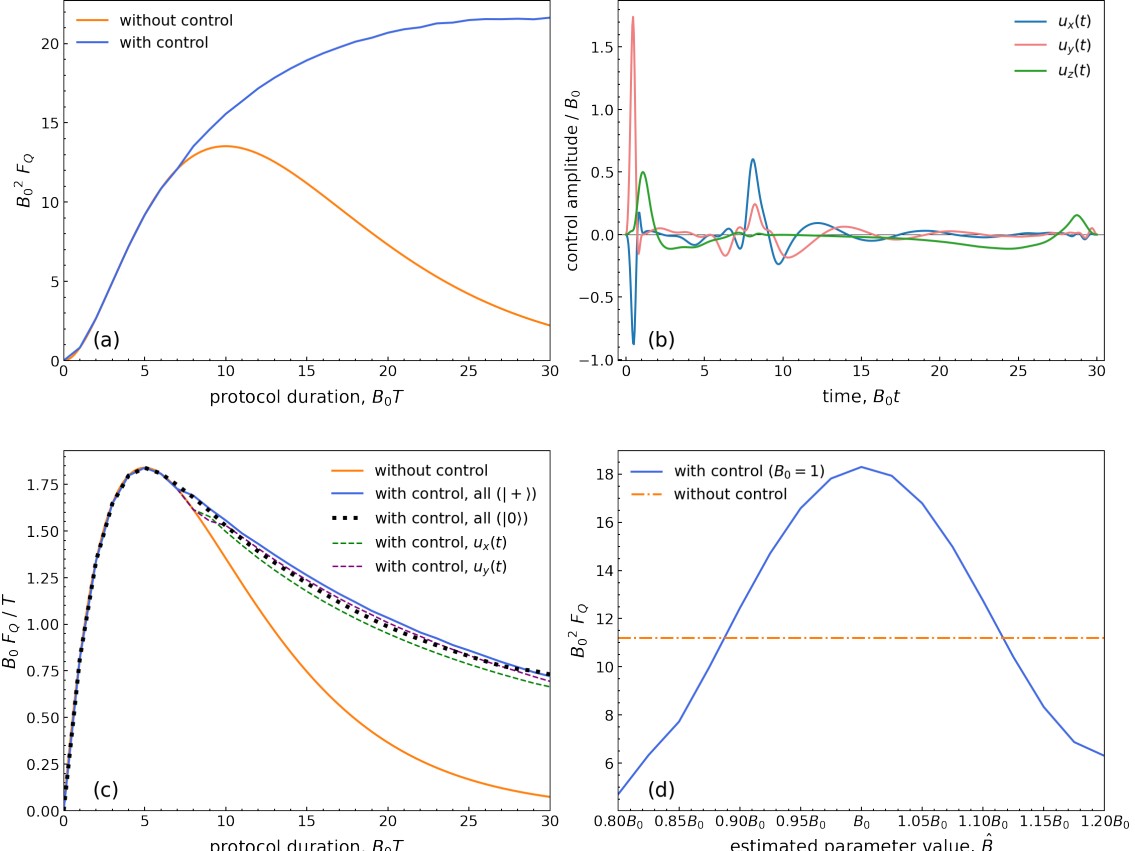

Figure 1: Parallel dephasing, $\gamma = 0.1$, $B_0 = 1.0$ (a) The QFI (in units of $B_0^{-2}$) achieved without control and with control optimised with different duration T (in units of $B_0^{-1}$). (b) Optimal controls obtained using Krotov's method for the dynamics with $T = 30$. (c) Normalised QFI (by T) without control (orange line), with control in all directions with $|+\rangle$ (purple line) or $|0\rangle$ (black dotted line) as initial state, and with control restricted to only $u_x(t)$ (green dashed line) or $u_y(t)$ (red dashed line). (d) The QFI achieved by applying controls optimised with different estimated parameter values $\hat{B}$ on states evolved under the true parameter value. The target time is $T = 15$. All axes are set to be dimensionless.

an absence of dissipative processes. The presence of noise drives both states toward the same steady state on a much shorter timescale set by the noise rate, which is independent of $\delta B$, thus significantly lowering the maximum reachable QFI [42].

## 3.1 Dephasing

Dephasing is a major and archetypical source of decoherence. It results in a loss of phase information with respect to a certain Hilbert space basis, represented as a shrinking of the $x$ and $y$ components of a state represented on the Bloch sphere (if the dephasing acts with respect to the $\sigma_z$ eigenbasis). Eventually, under dephasing, the initial state under slightly different system dynamics evolves towards steady states which are distinguished by a difference in their final projections along the $z$ axis. For dephasing parallel to the direction of the external magnetic field, the dynamics is described by (assuming $\hbar = 1$):

$$\dot{\rho} = -i\left[\mathcal{H}, \rho\right] + \frac{\gamma}{2}\left(\sigma_z \rho \sigma_z - \rho\right). \tag{12}$$

As neither $\mathcal{H}_0$ nor the parallel dephasing changes the $z$ projection of the initial state $|+\rangle$, the pair of states $\rho_B$ and $\rho_{B+\delta B}$ both shrink towards the centre of the Bloch sphere and cease to be distinguishable. In the Bloch representation, states can be expressed as $\rho(t) = \frac{1}{2}(\mathbb{I} + \vec{r}(t) \cdot \vec{\sigma})$ where $\vec{r}(t) = (r_1(t), r_2(t), r_3(t))$. The initial state $|+\rangle$ corresponds to $\vec{r}(0) = (1, 0, 0)$ and, without controls, evolves as $\vec{r}(t) = e^{-\gamma t}(cos(Bt), -sin(Bt), 0)$. The single-qubit QFI can be computed using the following formula [49]

$$F_Q(x) = |\partial_x \vec{r}|^2 + \frac{(\vec{r} \cdot \partial_x \vec{r})^2}{1 - |\vec{r}|^2} \; , \tag{13}$$

by setting the parameter of interest as $B$, resulting into a QFI without control under parallel dephasing given by $F_Q = T^2 e^{-2\gamma T}$. There is therefore a maximum at $T_{max} = 1/\gamma$, which corresponds to the coherence time, beyond which the effect of dephasing outweighs the fast dynamical separation between the two neighbouring states, reducing the distance between them, and correspondingly degrading the QFI.

As shown in Fig. 1(a), the QFI under parallel dephasing without control increases up to the estimation protocol duration $T_{max}$ then decreases, eventually approaching zero. In comparison, the QFI with optimal controls applied in all directions is able to increase beyond the coherence time. The maximum QFI attainable by the controlled dynamics has a higher value compared to that without control and is reached at a longer estimation duration $\left(T \approx 25 B_0^{-1}\right)$. The added controls allow for the QFI value to stabilise at the maximum value, and any further increase in the evolution time does not lead to further improvement. This shows that the states can be distinguished even after a long evolution, unlike under the uncontrolled dynamics where the evolution time must be comparable to the coherence time to ensure maximum QFI and thus the highest precision in parameter estimation.

When the dephasing effect on the system is less significant at shorter duration $T$ before $T_{max}$, the optimal strategy is to keep the probe state in the $x$-$y$ plane, where it is subject to the fastest dynamical separation. As such, the use of controls cannot improve the QFI. Once $T$ approaches and surpasses $T_{max}$, dephasing starts outweighing dynamical separation and, even on the $x - y$ plane, states away from the $z$ axis, which are most affected by the noise, lose distinguishability. The additional controls can now be activated, resulting in significant improvements in the QFI values attainable at these longer times.

The control pulse shown in Fig. 1(b) is representative of an entire class of solutions to the control problem of maximising the QFI at the end of the evolution. Fig. 2(a) shows the corresponding dynamics. In this case, as discussed in [42], the optimal controls exploit the decoherence free subspace of the states diagonal in the Pauli-$z$ eigenbasis (i.e., the $z$-axis of the Bloch sphere). An initial strong pulse in the y direction rotates the initial state close to the $z$-axis, where the effect of parallel dephasing is much reduced. The control fields are subsequently turned down to allow for the state to parameterise under free evolution, creating a separation between $\rho_B$ and $\rho_{B+\delta B}$, until the maximum rate of change in QFI is attained. Afterwards, the $x$ and $y$ control fields transfer the state back to near the $x$-$y$ plane, and the difference $\delta B$ results in one state to be just above the plane and one state to be just below, thus preserving the accumulated separation in the $z$ projection, unaffected by parallel dephasing. When the dephasing projects both states to the $z$ axis, the separation in the final $z$ projection is preserved indefinitely (since all states on the Bloch $z$ axis are steady states of the parallel dephasing dynamics), which explains the stabilisation towards a constant maximum QFI value observed in Fig. 1(a). Due to the states being away from the $x$-$y$ plane initially, the $z$ component of the control field is also activated. Individually, a control field in the $z$ direction would not be able to improve the QFI. The exact shape of the optimal control is dependent on the values of $B$, $T$ and $\gamma$. For example, a higher dephasing rate $\gamma$ requires a control with a stronger initial pulse to steer the state away from the $x$-$y$ plane as quickly as possible. A detailed analysis of

the impact of different decoherence rates on the maximum QFI attainable can be found in [42] (see, in particular, Fig. 5 therein).

To compare two schemes with the same cost of preparation and measurement, an important criterion for performance would be the gain in QFI per unit time (the 'normalised' QFI). For the uncontrolled scheme, the optimal duration for which increasing the duration of the estimation process leads to an increasing gain in QFI is up to $T_{opt} = 1/2\gamma$, after which further increase in $T$ has a decreasing yield. From Fig. 1(c), it can be observed that the use of controls does not improve the maximum value of normalised QFI, due to the gain from a higher maximum QFI attainable being cancelled out by the longer duration required, hence the time-efficiency of the controlled scheme remains the same. However, a comparison between variations of the controlled schemes in Fig. 1(c) shows that utilising optimal controls has other benefits in practical scenarios. The similar performance achieved by a controlled scheme using the $|+\rangle$ state and the $|0\rangle$ state under the optimal controlled schemes indicates that the estimation precision is not strongly dependent on starting with an optimal probe state. This happens because the first few iterations in time will converge to control fields which rotate any arbitrary initial state to the most suitable plane. Fig. 1(c) also shows that a single control field in the $y$ direction can reach nearly the same QFI scaling as the controls applied in all directions, which might be a more resource-efficient choice that does not compromise much on performance.

It is worth noting that in actual estimation problems the true value of the parameter $B$ is a priori unknown, so that an estimated value would be used to obtain an initial optimal control. Fig. 1(d) shows the QFI value obtained by controls optimised using different estimated $\hat{B}$, for the fixed true value $B_0 = 1$. It can be observed that the controlled scheme outperforms the uncontrolled scheme as long as $\hat{B} \in [0.9, 1.1]$, thus a controlled scheme is robust to an initial estimation error of up to 10%. Control fields can be obtained under a low-precision estimate and still improve the QFI towards obtaining a high-precision estimate, which can then be updated adaptively to further optimise the controls and approach the true value. The controls identified at the extrema of the tolerable parameter range ($\mp 10\%$ in the case above under dephasing) are still able to improve the QFI value obtained despite having a $\sim 5\%$ deviation

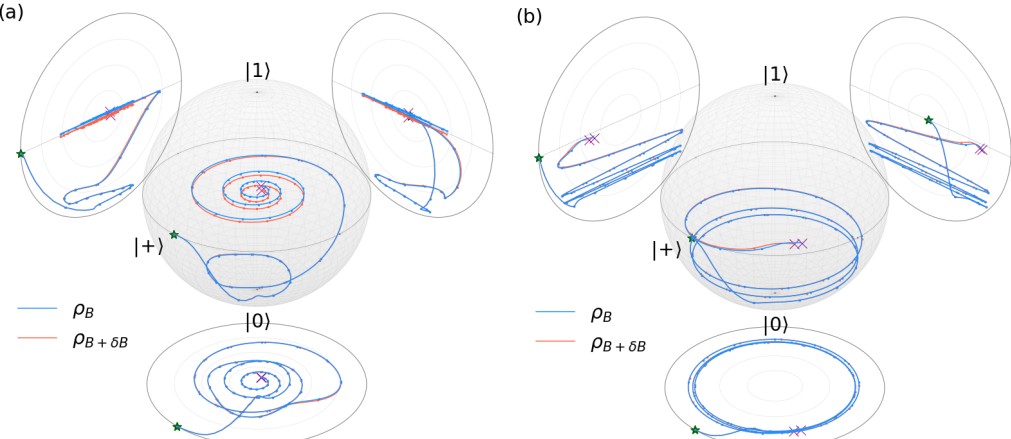

Figure 2: Controlled dynamics of the states $\rho_B$ and $\rho_{B+\delta B}$ displayed on the single-qubit Bloch sphere, for (a) parallel dephasing, subjected to the control fields shown in Fig. 1(b), and (b) relaxation, subjected to the control fields shown in Fig. 3(b). The green star and pink cross indicate the start and end of the evolution respectively. The density of dots reflects the speed of the evolution, with a low density indicating high speed.

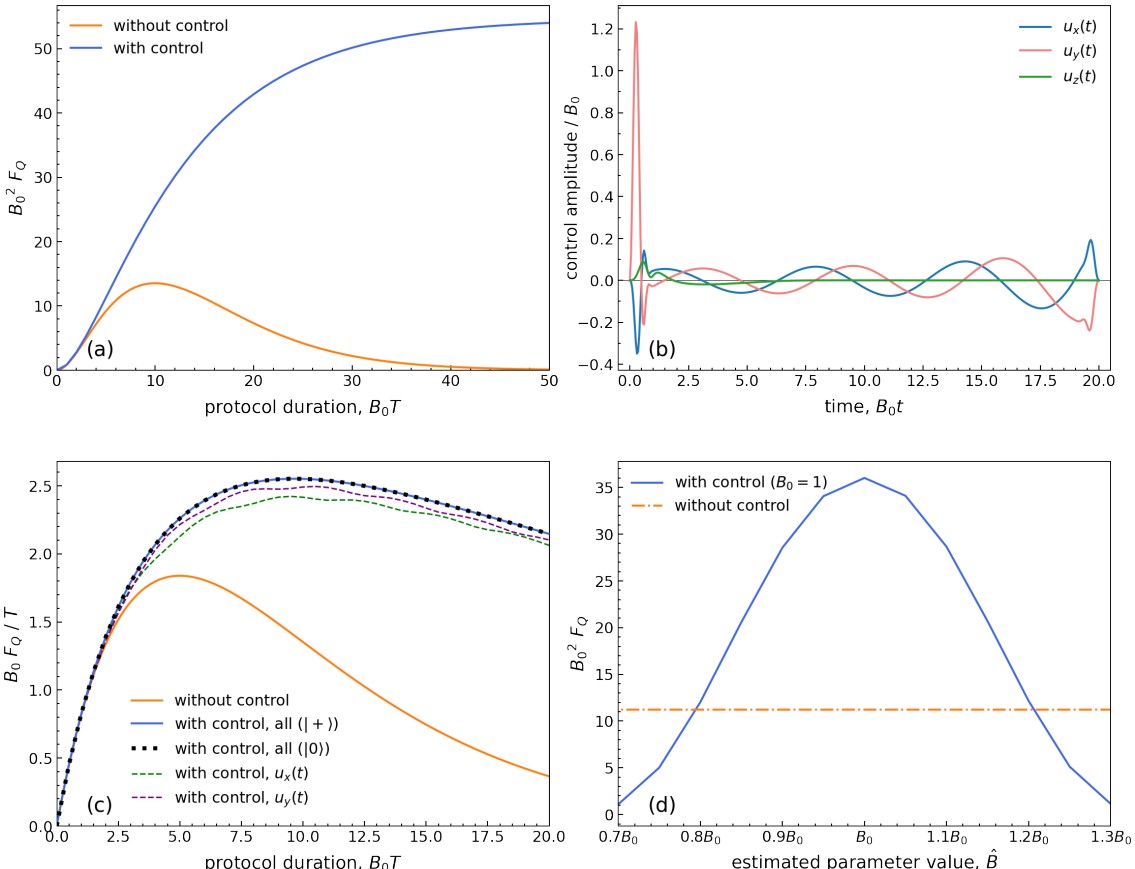

Figure 3: Relaxation, $\gamma_- = 0.2$, $B_0 = 1.0$ (a) The QFI (in units of $B_0^{-2}$) achieved without control and with control optimised at different duration $T$ (in units of $B_0^{-1}$). (b) Optimal controls obtained using Krotov's method for the dynamics with $T = 20$. (c) Normalised QFI (by $T$) without control (orange line), with control in all directions using $|+\rangle$ (purple line) or $|0\rangle$ (black dotted line) as probe and with control restricted to only $u_x(t)$ (green dashed line) or $u_y(t)$ (red dashed line) (d) The QFI achieved by applying controls optimised with different estimated parameter values $\hat{B}$ on states evolved under the true parameter value. The target time is $T = 15$.

from the ideal amplitudes (i.e., the control optimised at the true parameter value), as well as a $\sim$2% mismatch in the timing of the first amplitude peak. This shows that the controlled estimation scheme is fairly robust in the face of imperfections in the control pulses applied.

Let us also mention that the optimised control field is not a unique solution and does depend on the chosen method for numerical optimisation. In this regard, the controls obtained through Krotov's method in this study resemble a combination of the controls suggested in [42], which identified the initial rotation, and [32], which identified the need for an intermediate rotation as an optimal boost.

## 3.2 Relaxation

We further investigate the effects of optimal control for estimating the single parameter $B$ in $\mathcal{H}_0 = \frac{1}{2}B\sigma_z$ in the presence of relaxation by spontaneous emission from the excited state to the ground state, which is another major source of decoherence, for instance in atomic or linear

optics systems. In this case, the dynamics is described by:

$$\dot{\rho} = -i\,[\mathcal{H}, \rho] + \gamma_- \left( \sigma_- \rho \sigma_+ - \left\{ \frac{1}{2} \sigma_+ \sigma_-, \rho \right\} \right), \tag{14}$$

where $\sigma_- = (\sigma_x - i\sigma_y)/2$ is a lowering operator and $\gamma_-$ is the relaxation rate. This results in loss of energy for the excited states and evolves the initial $|+\rangle$ state towards the steady state $|0\rangle$.

In the Bloch representation, the initial state $|+\rangle$ evolves as:

$$r_1(t) = e^{-\frac{1}{2}\gamma_- t} \cos(Bt), \tag{15a}$$

$$r_2(t) = -e^{-\frac{1}{2}\gamma_- t} \sin(Bt), \tag{15b}$$

$$r_3(t) = e^{-\gamma_- t} - 1. \tag{15c}$$

Using Eq. (13), the QFI without control under relaxation scales as $F_Q = T^2 e^{-\gamma_- T}$, reaching a maximum at $T_{max} = 2/\gamma_-$. The effect of relaxation drives the final states reached for both $\rho_B$ and $\rho_{B+\delta B}$ towards the ground state where the magnetic field in the $z$ direction cannot affect them further, hence the final separation between them, and correspondingly the QFI, tends to zero.

The effects of applying optimal controls in this instance are shown in Fig. 3(a). Unlike in the previous case with parallel dephasing, there is a clear improvement in the attainable QFI with controls with respect to the uncontrolled dynamics, even at relatively short durations $T$. Also, the QFI converges to a much higher maximum value with the use of control, showing an improvement by a factor of five, and at a much longer duration ($T \approx 10 B_0^{-1}$). The QFI reaches a plateau at the maximum value, analogously to what was seen for dephasing, and stabilises over a timescale much longer than the coherence time of the system.

The general shape of the control field is similar to the one shown in Fig. 3(b) for all durations $T$, starting with a strong pulse in the $y$ direction to transfer states close to the ground state, then with subsequent smooth alternating fields in the $x$ and $y$ directions to evolve the states gradually towards the $x$-$y$ plane, where they separate most quickly and are not the most affected by the relaxation noise. The states are steered to experience fast dynamical separation by accumulating phase differences from rotations around the $z$ axis and to counteract decay to the ground state. The pulse applied at the end is a reverse of the initial pulse to transfer the states back to the $x$-$y$ plane, ensuring that state separation becomes maximum in the end. The corresponding dynamics is shown in Fig. 2(b). The initial $\pi/2$-like kick and its inverse counterpart at the end are similar to those identified for a control restricted to the $u_y(t)$ field in [42]. Under relaxation, the states are first steered towards the (decoherence free) ground state and then, after a few precessions around the $z$-axis, towards the $x$-$y$ plane. Although the protection from decoherence achieved through this trajectory is not as complete as in the case of dephasing, where a decoherence free subspace is used, states with different parameters separate faster than in the dephasing free subspace, resulting in a better distinguishability and thus a higher maximum QFI value (the reader should bear in mind that the QFI reflects the cumulative effect of dynamics upon the evolving state, rather than the information in the initial state, whose preservation is immaterial here).

If the controls are applied to maximise the QFI at a very long duration $T$, the states are kept around the ground state for most of the estimation process, to protect against relaxation. The alternating $x$ and $y$ fields are only activated towards the end of the system evolution, and the control scheme always ends with a rotation back to the $x$-$y$ plane. The control strategy can be scaled over an arbitrary duration, which accounts for being able to maintain the maximum QFI value at long durations.

As apparent from Fig. 3(c), the normalised QFI under the controlled scheme beats the uncontrolled value at estimation durations comparable to or longer than the coherence time

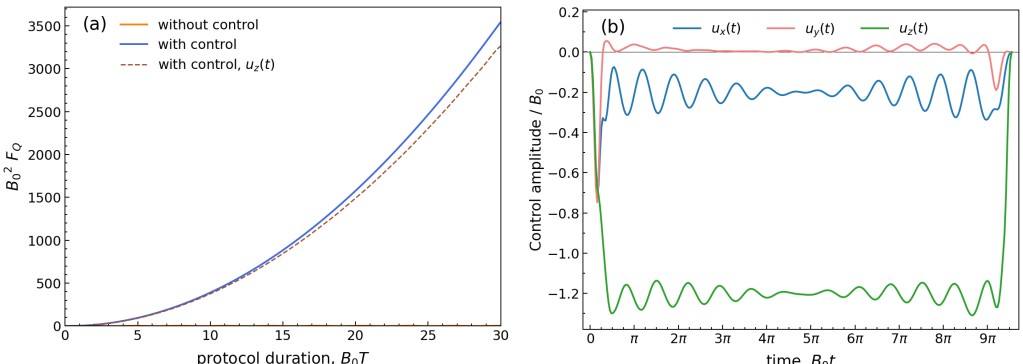

Figure 4: Non-commuting dynamics without noise (a) The QFI (in units of $B_0^{-2}$) achieved with controls applied in all directions (blue line) and restricted to only $u_z(t)$ (brown dashed line), as a function of duration $T$ (in units of $B_0^{-1}$), showing a near Heisenberg scaling (b) Optimal controls obtained using Krotov's method for the dynamics with $T = 30$.

$T_{opt} = 1/\gamma_-$, demonstrating the enhanced time-efficiency of the controlled estimation scheme. As should be expected, in this case too it is not necessary to prepare an optimal probe state to achieve optimal precision, since unconstrained unitary controls are capable to orient any probe state at leisure. The QFI scaling is indeed the same with all probe states, which was verified directly as a test for the numerical optimisation method by setting the initial state to $|+\rangle$ and $|0\rangle$ (Fig. 3(c)). Note that, however, applying controls in only the $y$ direction is also sufficient to approach the performance of controls in all directions, making it a viable alternative in practice. Moreover, the controlled scheme can improve QFI even if it is optimised under an estimated value deviating away from the true value by up to 20% (Fig. 3(d)), making it very robust to estimation errors.

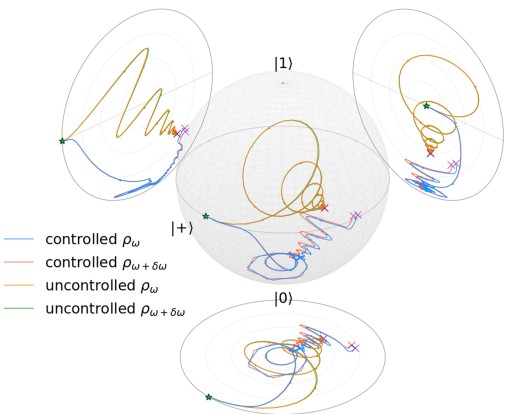

Figure 5: Controlled and uncontrolled dynamics of the states $\rho_\omega$ and $\rho_{\omega+\delta\omega}$ displayed on the single-qubit Bloch sphere. The control fields applied are shown in Fig. 6(b). The green star and pink cross indicate the start and end of the evolution respectively. The density of dots reflects the speed of the evolution, with a low density indicating high speed.

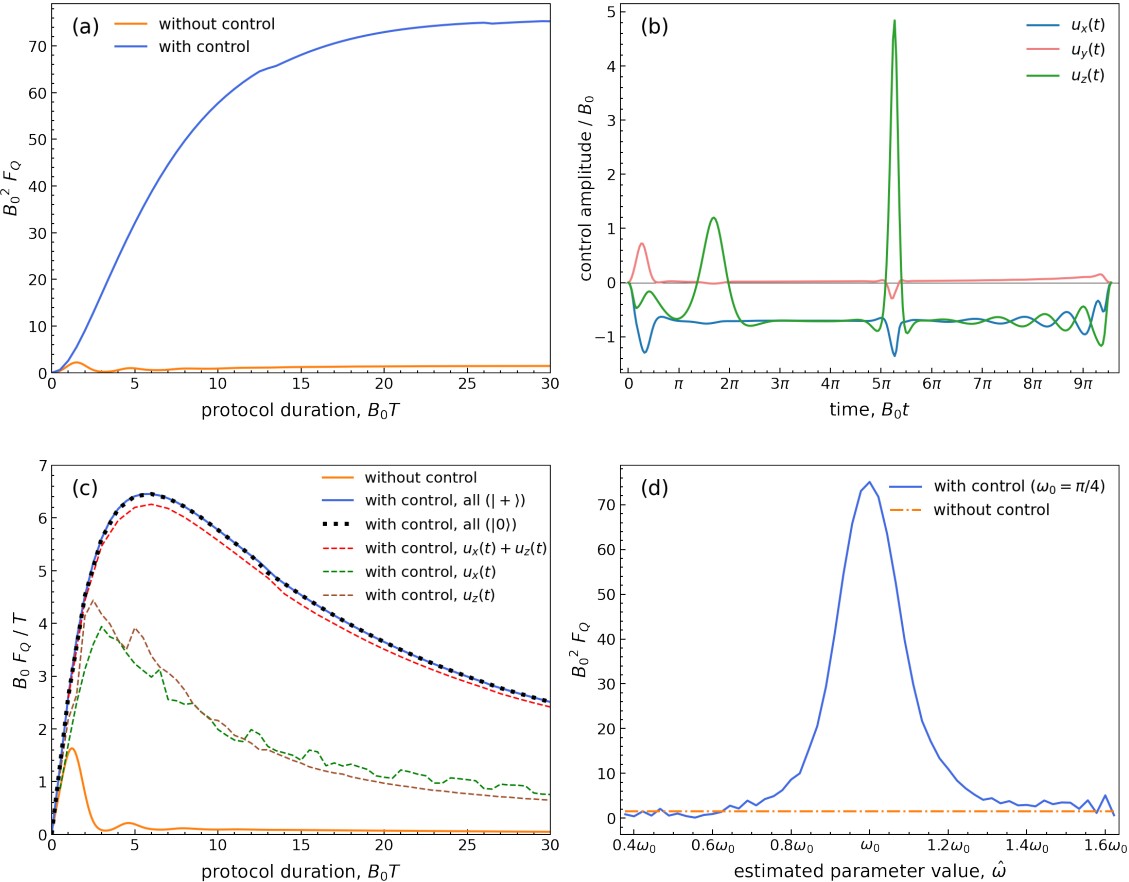

Figure 6: Non-commuting dynamics with parallel dephasing $\gamma = 0.1$ and relaxation $\gamma_- = 0.2$, $\omega_0 = \pi/4$ radian (a) The QFI (in units of $B_0^{-2}$) achieved without control, and with control optimised at different duration $T$ (in units of $B_0^{-1}$). (b) Optimal controls obtained using Krotov's method for the dynamics with $T = 30$. (c) Normalised QFI (by $T$) without control (orange line), with control in all directions (blue line), restricted to only $u_x(t)$ (green dashed line) or $u_z(t)$ (brown dashed line), and using a combination of both $u_x(t)$ and $u_z(t)$ (red dashed line). (d) The QFI achieved by applying controls optimised with different estimated parameter values $\hat{\omega}$ on states evolved under the true parameter value $\omega_0$.

## 3.3 Non-commuting dynamics

To address estimation in non-commuting dynamics, let us now set the magnitude of the magnetic field to unity, i.e. $B/2 = 1$ in Eq. (11), and consider the estimate of the parameter $\omega$ in the Hamiltonian $\mathcal{H}_0 = sin(\omega)\sigma_x + cos(\omega)\sigma_z$, which represents the direction of a magnetic field lying in the $x$-$z$ plane. The true parameter value is assumed to be $\omega_0$. In such a case, Hamiltonians at different values of $\omega$ do not commute with each other and estimation behaves very differently from the situations considered thus far. The QFI of this system in the noiseless and uncontrolled case is bound and oscillatory, with a time-scaling of $4\sin^2(T)$ [31], in contrast with the scaling for multiplicative Hamiltonians, which is unbound and monotonically increasing, and reaches the Heisenberg limit in the absence of noise.

In this case, sequential controls in the form of optimal rotations interspersed throughout the evolution have already been shown to improve the QFI attainable for such a system in the absence of noise [31]. The aim of the rotations is to eliminate the effect of non-vanishing commutators on the Bures distance between states with neighbouring parameters. Such uni-

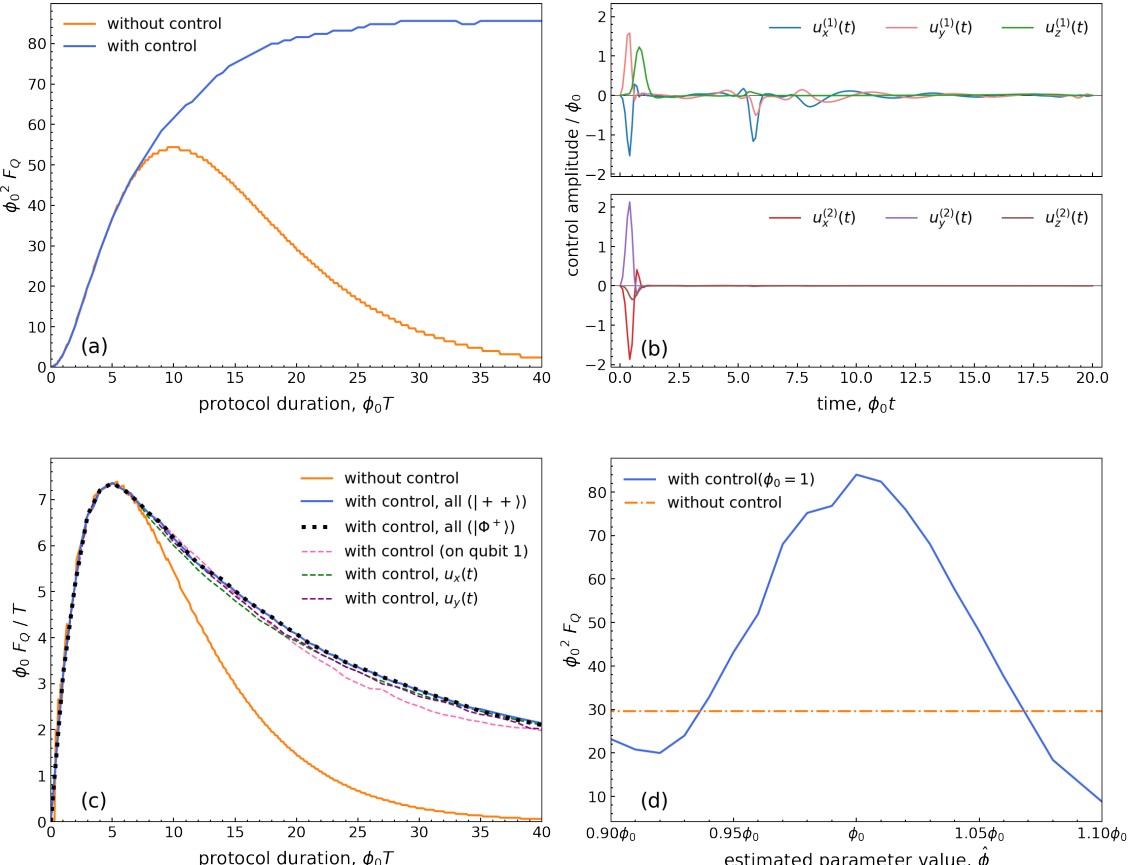

Figure 7: Local frequency estimation for a two-qubit system with ZZ coupling, $\phi_0 = 1.0$, $g = 0.1$ and $\gamma = 0.1$ (a) The QFI (in units of $\phi_0^{-2}$) achieved without control and with control optimised for target time $T$ (in units of $\phi_0^{-1}$). (b) Optimal controls obtained using Krotov's method for the dynamics with $T = 20$. (c) Normalised QFI (by $T$) without control (orange line), with control on both qubits using a product state $|++\rangle$ (blue line) or a maximally entangled state $\left|\Phi^+\right\rangle$ (black dotted line) as probes, applying controls in all directions only on the qubit with the local parameter of interest (pink dashed line), on both qubits restricted to $u_x(t)$ (green dashed line) or restricted to $u_y(t)$ (purple dashed line). (d) The QFI achieved at $T = 20$ by applying controls optimised with different estimated parameter values $\hat{\phi}$ on states evolved under the true parameter value $\phi_0$.

tary rotations can be applied at regular time intervals, separated by $t = T/N$, where $T$ is the overall duration of the control and $N$ is the number of rotations applied. The QFI then scales as $4N^2 sin^2(T/N)$. With $N$ controls, the maximal QFI that can be achieved is therefore $4N^2$ and the total time required to reach this value is $N\pi/2$, applying each rotation at an interval $t = \pi/2$. When $N$ is sufficiently large, the QFI approaches the optimal $4T^2$ scaling, but of course in practice the number of applicable controls is always limited to a finite amount. Hence the performance of this scheme may be limited in real experiments. An experimental implementation of this scheme using optimally timed $\sigma_z$ controls showed that while the estimation precision is able to surpass the shot-noise limit with just a few rotations, the effect of noise still sets a bound on the maximum precision and prevents it from attaining the Heisenberg scaling [50].

Here we show that, by applying a time-continuous optimal control field identified using Krotov's method, the Heisenberg scaling of $4T^2$ can be almost fully recovered, as shown in

Fig. 4(a). The optimal control has a relatively simple pulse shape, with a small constant negative offset by the $u_x(t)$ field and a large constant negative offset by the $u_z(t)$ field, on top of which there are slight modulations in all three control fields. The controls act to align the states to the direction of the external $x$-$z$ field, thus eliminating the effects of the non-commuting dynamics. The states are then rotated along an axis perpendicular to the external field, allowing for maximal accumulation of relative phase difference between the two neighbouring states, leading to fast separation and high distinguishability. The $u_z(t)$ field contributes most to the controlled estimation scheme, and restricting to only applying an optimal $u_z(t)$ control field was found to achieve a QFI scaling close to that of applying controls in all directions (Fig. 4(a)).

This continuous controlled scheme is also effective for the non-commuting dynamics subject to various sources of noise. The control was optimised for the system in the presence of both dephasing and relaxation. As shown in Fig. 6(a), the improvement in QFI significantly exceeds the bound on the uncontrolled scheme, approximately by a factor of ten. The oscillation in the QFI values is eliminated and approaches a constant maximum value, showcasing the control scheme's remarkable stability in time. The optimal control field identified (Fig. 6(b)) consists of an initial transfer of states close to the ground state, which is maintained by constant negative $u_x(t)$ and $u_z(t)$ fields that counteract the $x$ and $z$ terms in the system Hamiltonian as well as the effects of noise. A strong $u_z(t)$ pulse is applied at intervals to maintain the accumulated phase difference between the neighbouring states, thus enhancing distinguishability achieved in the long run. The corresponding dynamics is shown in Fig. 5.

Fig. 6(c) shows that the normalised QFI of the controlled scheme increases significantly with respect to the uncontrolled scheme, and the optimal control duration occurs early in the evolution, indicating excellent time-efficiency of the scheme. Individual control fields applied in the $x$ and $z$ directions can both improve the QFI attainable, but are significantly less effective than the controls applied in all directions. The control field applied requires at least two components, a combination of fields in the $x$ and $z$ directions, to reach close to the maximum attainable QFI, unlike in previous cases where control applied in a single direction would be sufficient. Fig. 6(d) shows that the identified controls beat the uncontrolled scheme over a broad range ($\sim$ 37.5%) of estimation errors, making it useful even if one can only obtain a very rough estimate of the parameter initially. This partial case study indicates that estimation under more general non-commuting dynamics does significantly benefit from applying additional Hamiltonian controls.

# 4 Application to two interacting qubits

Further to enhancing parameter estimation in a single-qubit system, we can also apply optimal control to the estimation of parameters in a two-qubit system. To that aim, we shall consider a system of two spin-1/2 subsystems, with a Hamiltonian given by:

$$\mathcal{H}_0 = \phi_1 \sigma_z^{(1)} + \phi_2 \sigma_z^{(2)} + g \mathcal{H}_{int} \ . \tag{16}$$

Estimation of the parameter of interest is considered in two cases: for ZZ coupling, i.e. $\mathcal{H}_{int} = \sigma_z^{(1)} \sigma_z^{(2)}$, and for XX coupling, i.e. $\mathcal{H}_{int} = \sigma_x^{(1)} \sigma_x^{(2)}$. The controls are applied locally on each qubit in all directions, $\mathcal{H}_c = \sum_{i=1,2} \vec{u}^{(i)}(t) \cdot \vec{\sigma}^{(i)}$. The optimal probe for the system with ZZ coupling is the $|++\rangle$ state, with associated QFI scaling of $4T^2$ for estimates of all three parameters $\phi_1$, $\phi_2$ and $g$ without noise. This is obtainable due to the fact that the three terms in $\mathcal{H}_0$ commute with each other and hence allow one to attain the Heisenberg scaling [51]. Under the presence of independent dephasing acting locally on each of the qubits,

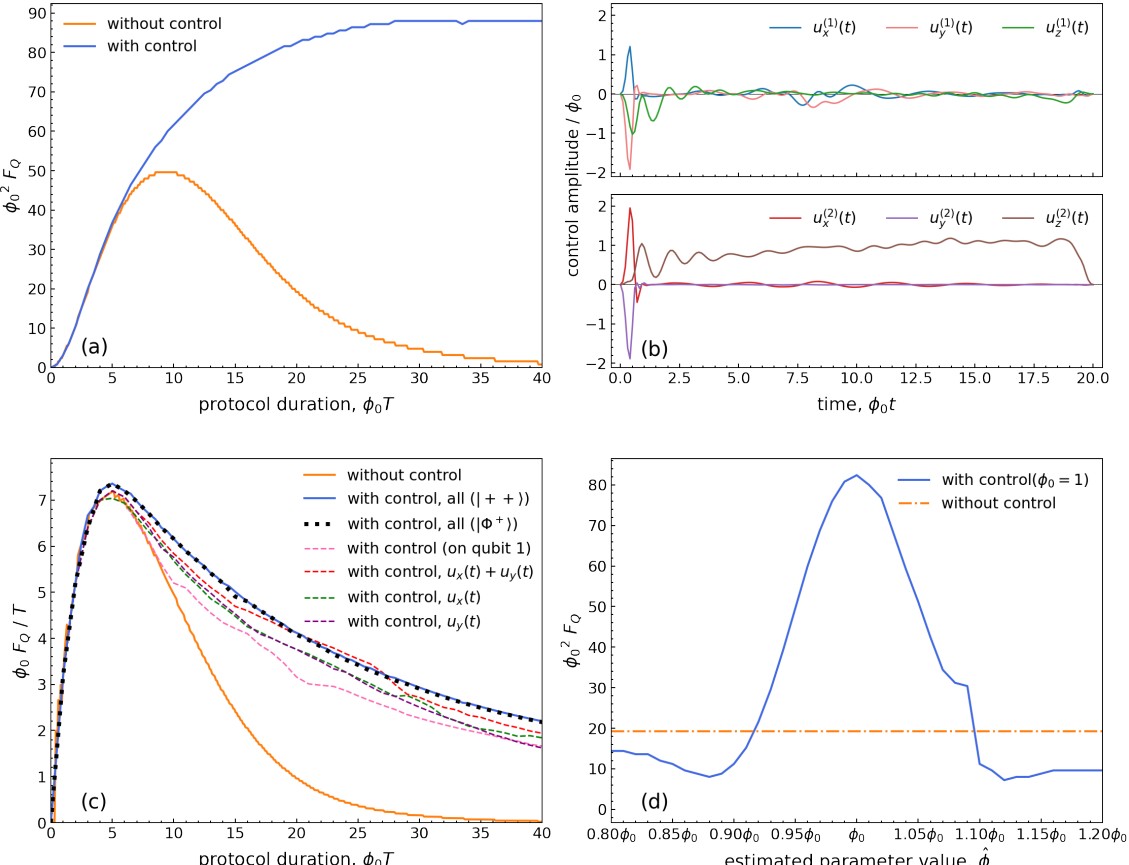

Figure 8: Local frequency estimation for a two-qubit system with XX coupling, $\phi_0 = 1.0$, $g = 0.1$, dephasing rate $\gamma = 0.1$ (a) The QFI (in units of $\phi_0^{-2}$) achieved without control and with control optimised for target time $T$ (in units of $\phi_0^{-1}$). (b) Optimal controls obtained using Krotov's method for the dynamics with $T = 20$. (c) Normalised QFI (by $T$) without control (orange line), with control on both qubits using a product state $|++\rangle$ (blue line) or a maximally entangled state $\left|\Phi^+\right\rangle$ (black dotted line) as probe, applying controls only on qubit 1 (pink dashed line), on both qubits restricted to only $u_x(t)$ (green dashed line), only $u_y(t)$ (purple dashed line) or having both $u_x(t) + u_y(t)$ (red dashed line). (d) The QFI achieved at $T = 20$ by applying controls optimised with different estimated parameter values $\hat{\phi}$ on states evolved under the true parameter value $\phi_0$.

the dynamics is described by:

$$\dot{\rho} = -i[\mathcal{H}, \rho] + \sum_{i=1, 2} \frac{\gamma_i}{2} \left(\sigma_z^{(i)} \rho \sigma_z^{(i)} - \rho\right), \tag{17}$$

where $\gamma_i$ is the dephasing rate for the $i$th qubit. The dephasing on the two qubits results in a mixed dephasing time $\tau$, which is generally not shorter than the local dephasing on each of the qubits, i.e. $\tau \geq \tau_1, \tau_2$, where $\tau_i = 1/\gamma_i$. Hence the overall decoherence is still defined by local dephasing [52]. For the purpose of this study, the dephasing rate is assumed to be the same for both qubits, i.e. $\gamma_1 = \gamma_2 = \gamma$.

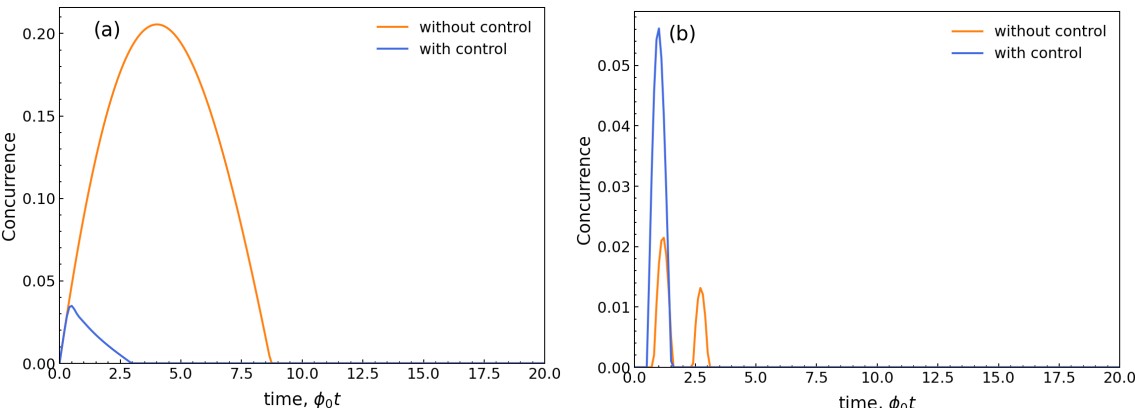

Figure 9: Concurrence of uncontrolled and optimally controlled (for local phase estimation) two-qubit systems under ZZ (a) and XX (b) couplings. The probe state is $|++\rangle$.

## 4.1 Estimating a local frequency

We first consider the estimate of a local frequency of one of the two qubits (qubit 1) for the system with ZZ coupling, $\mathcal{H}_{int} = \sigma_z^{(1)}\sigma_z^{(2)}$. Let $\phi_1 = \phi_2 = \phi$, and let $\phi$ be the parameter of interest to optimise for, with true value $\phi_0$. Fig. 7(a) shows that the QFI of the two-qubit system under dephasing without control reaches a maximum at the duration $T_{max} = 1/\gamma$ (the coherence time of the individual qubit), while the controlled QFI increases beyond the coherence time and stabilises towards a constant maximum value. The extent of the increase in QFI is comparable to the single qubit dephasing case, in which the QFI values begin to improve when the duration of the system evolution is close to the coherence time, and increases further until stabilising at a longer duration ($T \approx 30$). Fig. 7(b) presents an example of the optimised control fields when control is applied in all directions. The optimised control field acts on qubit 1 by implementing an initial rotation close to the ground state along with intermediate 'boosts' in the $x$ and $y$ directions, around $T_{max}$, back to the $x$-$y$ plane, much like for the optimal controls identified for the single-qubit system under dephasing. The qubit without the local parameter of interest is driven to the ground state and remains there throughout.

From the plot of the normalised QFI shown in Fig. 7(c), it may be seen that the controlled estimation scheme does not improve the overall time efficiency of the estimation process, in that a time of the order of $1/\gamma$ is required to achieve significant increase in QFI. The estimation is also largely independent from the initial probe states, even with regard to the presence of initial entanglement, which is the case since controls act to quickly disentangle the system, thus purifying the qubit under estimation. This behaviour is confirmed by a direct analysis of the entanglement between the two qubits during the controlled evolution, illustrated in Fig. 9, where the concurrence $C$, a convenient quantifier of two-qubit entanglement [53], varying from $C = 0$ for a disentangled state to $C = 1$ for a maximally entangled state, is plotted. In addition to applying full control in all directions, several alternative ways to implement control are compared in Fig. 7(c). Like in single-qubit case, restricting the controls applied to only the $x$ or $y$ direction on both qubits achieves a similar performance as having controls in all directions. Controlling only the qubit which corresponds to the parameter of interest also achieves a good performance, which, along with the similarity with the single-qubit scenarios, is most likely due to fact that the interaction term commutes with the local term to be estimated. At variance with the single-qubit case, the tolerance to estimation error when optimising the control for this two-qubit system holds only over a small range ($\sim 5\%$) (Fig. 7(d)), in part due to the larger and more sensitive control landscape: the initial estimate of the parameter must be

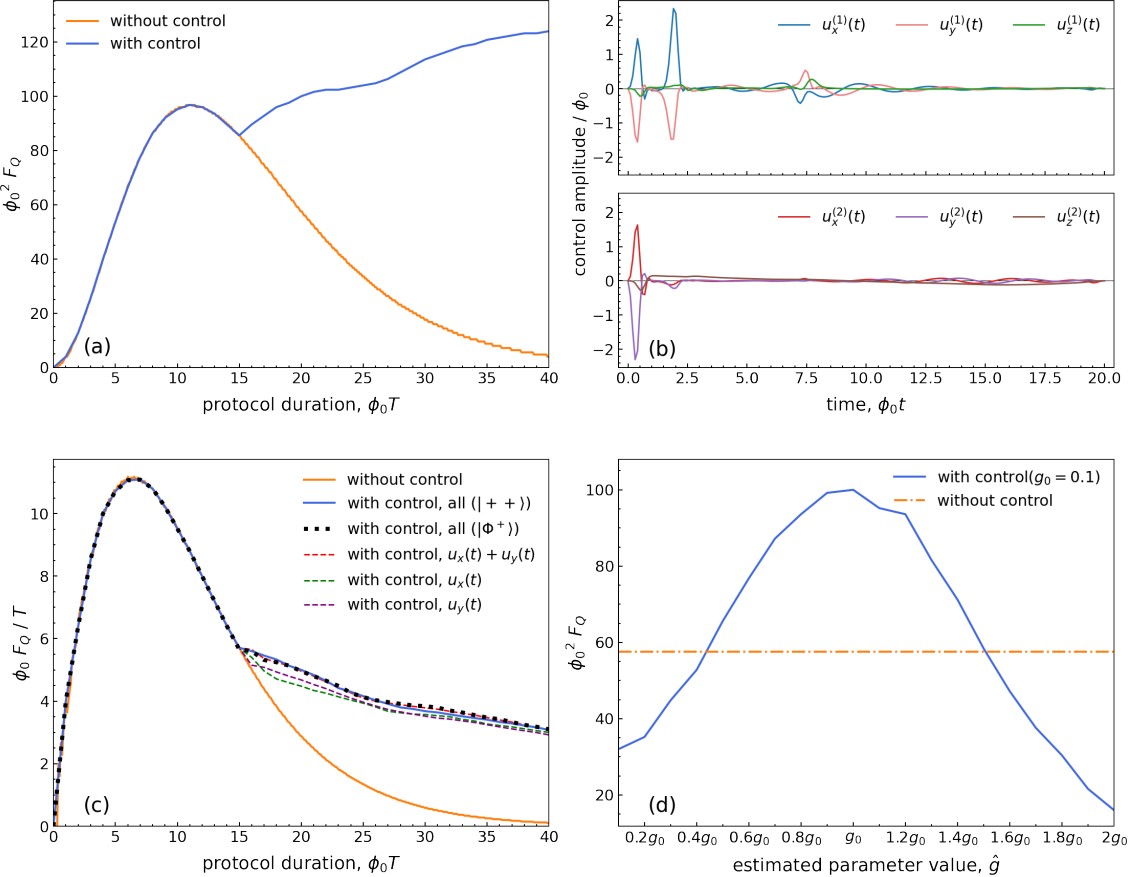

Figure 10: Interaction strength estimation for a two-qubit system with ZZ coupling, $g_0 = 0.1$, $\phi = 1.0$, dephasing rate $\gamma = 0.1$ (a) The QFI (in units of $\phi_0^{-2}$) achieved without control and with control optimised over a duration $T$ (in units of $\phi_0^{-1}$). (b) Optimal controls obtained using Krotov's method for the dynamics with $T = 20$. (c) Normalised QFI (by $T$) without control (orange line), with control on both qubits using a product state $|++\rangle$ (blue line) or Bell state $\left|\Phi^+\right\rangle$ (black dotted line) as probe, with control restricted to $u_x(t) + u_y(t)$ (red dashed line) on both qubits, restricted to $u_x(t)$ (green dashed line) or restricted to $u_y(t)$ (purple dashed line). (d) The QFI achieved at $T = 20$ by applying controls optimised with different estimated parameter values $\hat{g}$ on states evolved under the true parameter value $g_0$.

relatively accurate for the optimisation to identify the best control fields. The capability of controlled estimation to achieve an improved QFI value under a mismatch in control fields applied, compared to 'ideal' control fields optimised at the true parameter value, also reflects a degree of tolerance to the use of imperfect control fields. In this case of frequency estimation under dephasing with two ZZ-coupled qubits, the controls identified at the extrema of the tolerable parameter range ($\mp 5\%$ of the true parameter value) have a $\sim 5\%$ deviation from the ideal amplitudes and a $\sim 1.5\%$ mismatch in the timing of the first amplitude peak. While less tolerant to a mismatch between the estimate used for optimisation and the true parameter value, the controlled estimation scheme remains robust to imperfections in the control pulses applied.

In the case of having a transverse XX coupling instead of ZZ coupling, where $\mathcal{H}_{int} = \sigma_x^{(1)}\sigma_x^{(2)}$, it can be seen from Fig. 8(c) that the uncontrolled QFI of the local frequency is lower than the ZZ coupling one, while applying optimised controls allows one to recover the same QFI scaling. The optimal pulse identified (Fig. 8(b)) rotates the state of the qubit

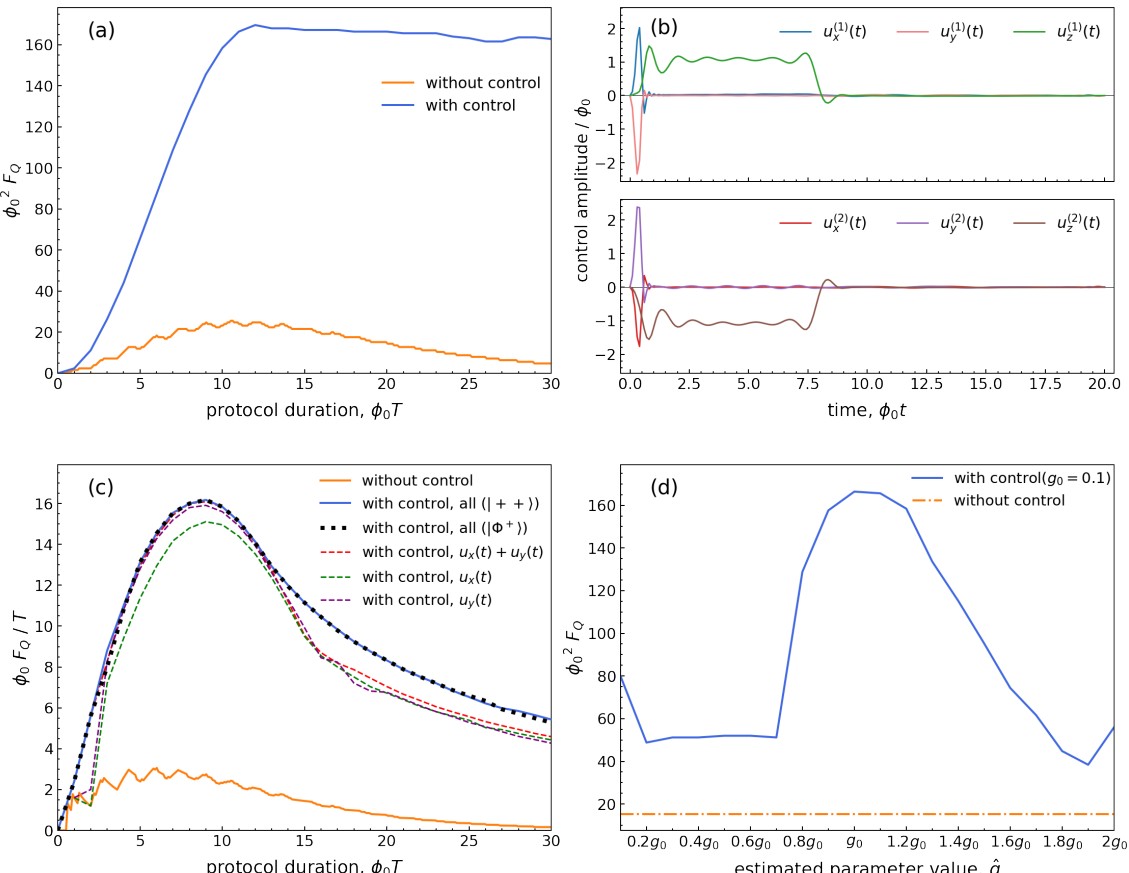

Figure 11: Interaction strength estimation for a two-qubit system with XX coupling, $g_0 = 0.1$, $\phi = 1.0$, dephasing rate $\gamma = 0.1$ (a) The QFI (in units of $\phi_0^{-2}$) achieved without control and with control optimised over target time $T$ (in units of $\phi_0^{-1}$). (b) Optimal controls obtained using Krotov's method for the dynamics with $T = 20$. (c) Normalised QFI (by $T$) without control (orange line), with control on both qubits using a product state $|++\rangle$ (blue line) or Bell state $\left|\Phi^+\right\rangle$ (black dotted line) as probe, with control restricted to $u_x(t) + u_y(t)$ (red dashed line) on both qubits, restricted to $u_x(t)$ (green dashed line) or restricted to $u_y(t)$ (purple dashed line). (d) The QFI achieved at $T = 20$ by applying controls optimised with different estimated parameter values $\hat{g}$ on states evolved under the true parameter value $g_0$.

containing the parameter of interest close to $+z$ direction and lets it rotate naturally about the $z$ axis while falling back to the $x$-$y$ plane. The state of the other qubit is aligned to the $z$ axis and maintained there by having a constant $u_z(t)$ field. The controlled estimation scheme identified for this system has a slightly greater tolerance to estimation errors in the values initially used for identifying the optimal control, compared to the case with ZZ coupling (Fig. 8(d)). There is a guaranteed gain in QFI over a range of close to a 10% underestimation or overestimation in the initial value of the frequency $\phi$.

We also consider several alternative implementations of controls in Fig. 8(c). Unlike the previous case, applying control fields only in the $x$ and/or $y$ direction, or only on a single qubit does not allow one to achieve a similar performance to applying the full control. For this case with XX coupling, the control must at least be applied in two directions on both qubits to have sufficient control over the system dynamics and reach close to the maximally attainable QFI.

## 4.2 Estimating the interaction strength

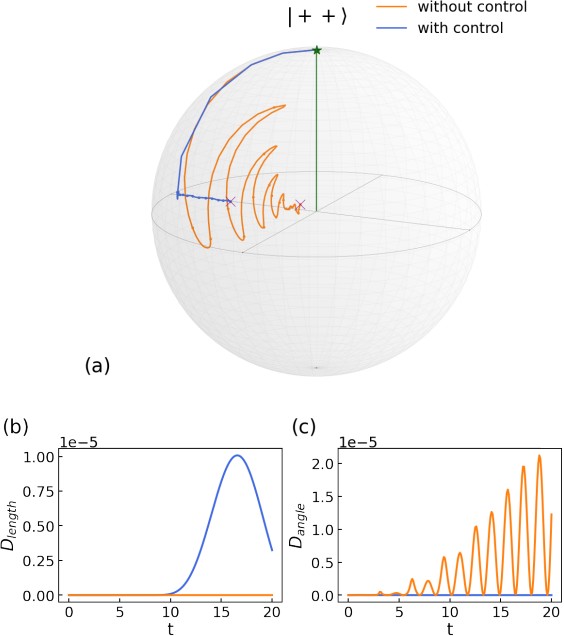

(a)

(b)                    (c)

Figure 12: (a) Generalised Bloch vectors showing the evolution of the two-qubit states for interaction strength estimation with XX coupling. The $|++\rangle$ probe state is aligned to the $+z$ direction as a reference state. The optimised control fields applied are shown in Fig. 11(b). The green star and pink cross indicate the start and end of the evolution respectively. The (b) difference in length and (c) angle between the Bloch vectors of the states $\rho_g$ and $\rho_{g+\delta g}$ are plotted for the controlled and uncontrolled dynamics

In this section, we consider the estimation of the parameter $g$, letting $\phi_1, \phi_2 = 1$, first for the case of $\mathcal{H}_{int} = \sigma_z^{(1)}\sigma_z^{(2)}$. The true parameter value is taken to be $g_0$. In this instance, the controlled estimation scheme allows for the attainable QFI to increase beyond the coherence time and stabilise towards a constant maximum value, as shown in Fig. 10(a). However, the extent of increase in the maximum QFI value attainable is relatively modest compared to all previous cases. Quite remarkably, for the configuration considered in Fig. 10(a), the con-

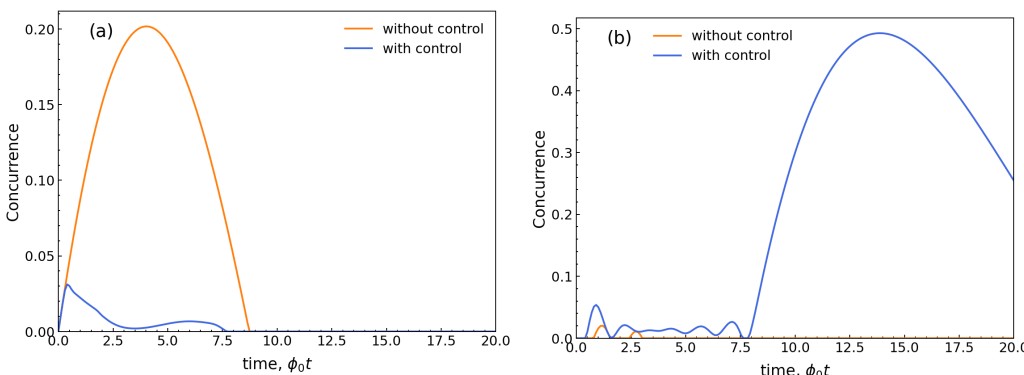

Figure 13: The evolution of concurrence of the states evolved with (a) ZZ coupling and (b) XX coupling, in presence of local dephasing, with and without control. The probe state is $|++\rangle$.

trolled QFI only beats the uncontrolled one for target times $T \gtrsim 15.5\phi_0^{-1}$ (hence the apparent discontinuity in the first derivative of the controlled QFI, since its value for $T < 15.5\phi_0^{-1}$ is replaced by the higher, uncontrolled one in the plot). The controls cannot improve the QFI for a duration $T$ up to the coherence time, and only play a significant part once dephasing effects become substantial. The control fields shown in Fig. 10(b) are representative of the optimal fields over all durations $T > T_{max}$. On one qubit, there occur a few initial strong pulses aligning the state of the qubit diagonally on the Bloch sphere, followed by a weak intermediate pulse rotating the state back to the $x$-$y$ plane. The other qubit experiences a similar strong pulse at the start of evolution steering it close to the z axis, then the control subsides and the qubit rotates naturally about such an axis.

These two-qubit dynamics can be analysed in terms of 15-dimensional generalised Bloch vectors (Fig. 12(a)), formed by the state's components in a Hilbert-Schmidt orthogonal basis of $4{\times}4$ Hermitian matrices, whose Euclidean length determines the states' linear entropies. Such an analysis reveals that the improved distinguishability of the controlled states for neighbouring values of the estimated parameter is largely due to the fact that, for the controlled dynamics, the length of the Bloch vector (and thus the global state entropy) is very sensitive to the parameter. The Bloch vectors of the pair of neighbouring states become distinctively different in length under the controlled dynamics, resulting in a greater distance between them at the end of the estimation protocol, as shown in Fig. 12(b). This is in contrast to the uncontrolled case, where the lengths of the Bloch vectors stay virtually equal and most of the distance between the neighbouring states is down to an angle between their Bloch vectors, as shown in Fig. 12(c). This feature is common to local frequency estimation too, and is compounded by the fact that, in all of these case studies, the controls act generally to reduce the global state's entropy and preserve its coherence in the face of noise.

Fig. 10(c) shows that the normalised QFI does not improve under the controlled scheme since the gain in QFI is only obtainable at long estimation durations, thus the time-efficiency remains unchanged. The performance of the controlled scheme shows the same independence from using an optimal probe state as in previous cases. The QFI scaling achieved by applying control restricted to a single $x$ or $y$ direction on both qubits is very close to the performance of the full control, suggesting that it is not necessary for the control to be maximised over all degrees of freedom to reap the benefit of the improved QFI scaling in practice. However, all controlled schemes for this system are common in that they only show a gain over uncontrolled estimation at long durations. In this case, the tolerance to estimation error when optimising control fields is relatively large at $\sim 60\%$ (Fig. 10(d)), thus the control can be useful even if it is initially optimised with a grossly inaccurate estimate on the parameter.

For the case of $\mathcal{H}_{int} = \sigma_x^{(1)} \sigma_x^{(2)}$, the *uncontrolled* QFI attainable for an estimate of the parameter $g$ is much less than in the previous case with ZZ coupling, as a result of the non-commutativity arising from the transverse interaction term (Fig. 11(a)). In this case, the use of Hamiltonian controls results in a significant increase in the QFI values attainable, close to a factor of seven. The improved QFI scaling stabilises close to the maximum QFI attainable over a short duration of $T \simeq 10\phi_0^{-1} = \gamma^{-1}$, hence the gain in time efficiency of the controlled estimation scheme is also significant. Fig. 11(b) shows an example of the full optimal control identified, acting in all directions to keep one of the qubits in the ground state and the other one in the exact opposite excited state. Comparing the performance of several alternatives to implementing full control, the QFI scaling obtained from applying control fields restricted to the $x$ and $y$ directions, or a single $y$ direction, on both qubits is able to reach the same performance as applying full control, for durations up to $T \simeq 13\phi_0^{-1} = 1.3\gamma^{-1}$ (Fig. 11(c)). If the controlled scheme is implemented for a longer duration, full control is required to reach optimality. The controlled estimation scheme identified for this system is also very tolerant to estimation errors.

The temporal evolution of quantum entanglement during estimations in bipartite systems is also worth investigating. In Fig. 13, we report the evolution of the concurrence for uncontrolled systems and for systems optimally controlled towards interaction strength estimation, in both the ZZ and XX coupling cases. The presence of dephasing results in fast decaying off-diagonal terms, which eventually leaves a diagonal density matrix describing a merely classical probability distribution. Therefore, in the presence of noise, concurrence is expected to decay to zero on a timescale determined by the local dephasing time on each of the qubits [52].

It can be seen that, in estimating ZZ couplings, the controls tend to suppress the entanglement at intermediate times to maintain a maximal final estimation precision. Thus, it would appear that in this instance the Bures distance between evolving states with neighbouring interaction strengths is maximised when acting on separable states. However, in the case of XX coupling, we observe the converse behaviour whereby control enhances entanglement in establishing a maximal estimation precision at the end of system evolution. In this case, closer inspection of the system state shows that the entanglement starts to grow once the system approaches the product state $|01\rangle$: the following dynamics under transverse XX coupling happens to be both entangling and rapid (in terms of Bures distance) but, if contrasted with the ZZ coupling case, this coincidence seems to be circumstantial. Notice that, under general controls, which can offset any local Hamiltonian terms, the difference in estimation performance between ZZ and XX couplings must be entirely ascribed to the interplay between such interaction Hamiltonian terms and the dephasing noise: Quantum systems turns out to be more sensitive to couplings which are transverse with respect to the dephasing basis.

# 5  Conclusions

In summary, we have studied how the method of quantum optimal control can improve the precision of a single parameter estimation for various parameters of interest in single-qubit and two-qubit systems subjected to noise. Our results show that the use of quantum optimal control, in particular the adoption of Krotov's method, not only allows one to improve the achievable precision for parameter estimation in a simple single-qubit system with commuting dynamics, as already found in previous studies [32, 42], but is also applicable to single-qubit systems with more general non-commuting dynamics, as well as interacting two-qubit systems. Most remarkably, we could demonstrate the restoring of the Heisenberg limit in the non-commuting estimate of the field direction for a single noisy qubit, as well as a sevenfold advantage in terms of QFI over uncontrolled systems in the case of interaction strength estimation for a transverse coupling between two qubits.

Apart from gaining a higher estimation precision, the controlled schemes showed other desirable traits, such as the time stability in maintaining the maximum QFI over long time spans, observed consistently across all the examples investigated. The choice of an optimal probe state also becomes irrelevant as the optimised controls will orient an arbitrary probe state to the right plane at the start of the estimation protocol. In most cases, the relatively simple pulse shape identified, the possibility of constraining the pulses to single directions whilst maintaining near optimal performance and the tolerance to initial estimation errors all point to potential for practical impact.

Our quantitative analysis of the protocol's robustness, illustrated by the (d) plots of Figs. 1, 2, 4, 6, 7, 10, 11, shows that deviations from the ideal optimal controls still allow one to improve the estimation precision limit compared to uncontrolled estimation. The probe states can still be steered towards the decoherence free subspaces, so that controlled schemes have significant potential to outperform uncontrolled ones, even when faced with imperfections in the control field pulses' amplitudes and timings. The requirements on the precision of the

control pulses indicated by the case of frequency estimation under dephasing for a single-qubit and two ZZ-coupled qubits, which are the least tolerant to imperfect controls among the single-qubit and two-qubit cases respectively, are reasonable in view of current pulse shaping technology. In an actual experiment, one can also run the full protocol for a few values of the parameter value to locate the position of the peak performance. Notice also that, in extracting information about the parameter, the advantage of controlled dynamics over repeated uncontrolled ones would ultimately depend on the latter's cost in specific circumstances (although of course controls and repetition can be combined, attaining better information rates).

The present study offers several opportunities for developments and future research. For one thing, one could apply optimal quantum control methods to non-Hamiltonian parameters, such as the decoherence rates. There is also much to explore as to estimating parameters in more general two-qubit systems, subjected to other couplings or noise, or beyond the two-party scenario, extending it to many-body systems [54]. These complex systems naturally lead to the task of enhancing multiparameter estimation, currently a very lively area of research (for a systematic optimised approach to multiparameter Hamiltonian estimation, see Sec.III.E of [55]). The relation between disentangling effects of the controls and their effect on measurements and estimation might present another line of future inquiry.

## Acknowledgments

We thank Michael Biercuk for discussions and Matthias Krauß for his technical support. C.P. Koch acknowledges financial support from the Federal State of Hesse, Germany, through the SMolBits project within the LOEWE program and from the Einstein Research Unit on near-term quantum devices.

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
