# Peer review of "Optimal control for Hamiltonian parameter estimation in non-commuting and bipartite quantum dynamics"

_SciPost Physics, doi:SciPost Phys. 13, 121 (2022)_

## Round 1 · Referee Report · Anonymous (Referee 1) · 2022-6-2

Strengths
Unexpected and interesting results
Weaknesses
Hand wavy explanations on the numerical findings
Poor presentation of the results
Poor presentation of the results
Report
The authors explored the problem of maximizing the quantum Fisher information under decoherence. Their findings are quite interesting and maybe useful in practice. However, I cannot recommend it for publication in SciPost, at least in the current form.
The authors found that the use of quantum control drastically improved the maximum attainable QFI with T1 decoherence, in comparison to the case of T2. This is unexpected, and one would normally expect the opposite. The reason is that T1 washes away information contained in the Pauli-Z expectation value in addition to Pauli- X and Y. Therefore, it is harder to hide quantum information from T1 decoherence with control. I found the authors’ explanation on this surprising result unsatisfactory.
It will be useful to show that the scheme is robust to the unknown parameters in the drive Hamiltonian. Do small changes in B cause big changes in the controls?
Finally, I have a few suggestions on presentations: 1. Consider plotting the Bloch vectors (and their derivatives with respect to B) as functions of time. 2. A reader might be curious to know the maximum attainable QFI as a function of the decoherence rates. 3. Include some arguments on why one should use quantum control instead of just repeating the experiments for more times with reduced evolution times.
The authors found that the use of quantum control drastically improved the maximum attainable QFI with T1 decoherence, in comparison to the case of T2. This is unexpected, and one would normally expect the opposite. The reason is that T1 washes away information contained in the Pauli-Z expectation value in addition to Pauli- X and Y. Therefore, it is harder to hide quantum information from T1 decoherence with control. I found the authors’ explanation on this surprising result unsatisfactory.
It will be useful to show that the scheme is robust to the unknown parameters in the drive Hamiltonian. Do small changes in B cause big changes in the controls?
Finally, I have a few suggestions on presentations: 1. Consider plotting the Bloch vectors (and their derivatives with respect to B) as functions of time. 2. A reader might be curious to know the maximum attainable QFI as a function of the decoherence rates. 3. Include some arguments on why one should use quantum control instead of just repeating the experiments for more times with reduced evolution times.

---

## Round 1 · Referee Report · Anonymous (Referee 2) · 2022-6-12

Report
In Optimal control Hamiltonian parameter estimation [...] Qin et al. propose a method to determine what control fields to apply to a system in order to best estimate parameters of the Hamiltonian describing that system.
The concept of the paper is a creative application of optimal control and leads to a counterintuitive result. I have a concern over the validity of the results related to taking the effect of errors in the controlling fields into account (detailed below). Until this concern is addressed I cannot recommend this paper for publication.
After introducing the method, the authors work through two examples, the estimation of the magnetic field (frequency) of a single-qubit system, and the estimation of the coupling strength (both XX and ZZ) of a two-qubit system.
In the single-qubit example, the authors show the counterintuitive result that placing the qubit in the $|0\rangle$ state during most of the experiment and applying the $u_x$ and $u_y$ fields near the end of the estimation period results in a better estimate of the eigenfrequency of the system than placing the system in the $|+\rangle$ state during the entire estimation period as is commonly done in conventional Ramsey type estimation of the frequency.
One assumption the authors seem to make implicitly is that the controls $u_x$, $u_y$ and $u_z$ are ideal. In practice, the quantity that is being estimated (the frequency) is exactly the same quantity that is used to determine the phases $x$ and $y$ of the control fields $u_x$ and $u_y$. As such, I would expect the method to work only because the phase tracking that is normally done by the qubit state in the $|+\rangle$ state to now be done by the "ideal" control fields.
If the authors could explain intuitively how this is not the case, and the error in the control would not affect the estimate it would improve the manuscript significantly.
Although the 2-qubit example is not directly about estimating the qubit frequencies but rather the coupling, it is not clear that this estimate is not also affected by frequency errors in the control fields.
Some minor points.
When reading the abstract and introduction it was not immediately clear that this manuscript is about using optimal control to determine what fields to apply to acquire better estimates rather than using optimal control to determine what driving fields would optimally implement a certain operation. This is mostly because finding optimal control pulses is a very common application these days. Being a bit more explicit could help the casual reader identify what this paper is about quicker.
The images of the driving fields could be improved by adding a few images of the Bloch sphere depicting the (ideal) state of the qubit during different parts of the protocol. This would also help the intuitive understanding.
In the examples it is not clear to me if there is a discrete control pulse near the end of the estimation period $T$ or if the fields are brought to oscillate during the entire estimation period. The textual description seems to suggest the former while the figures (typically figure b) seem to suggest the latter.
The concept of the paper is a creative application of optimal control and leads to a counterintuitive result. I have a concern over the validity of the results related to taking the effect of errors in the controlling fields into account (detailed below). Until this concern is addressed I cannot recommend this paper for publication.
After introducing the method, the authors work through two examples, the estimation of the magnetic field (frequency) of a single-qubit system, and the estimation of the coupling strength (both XX and ZZ) of a two-qubit system.
In the single-qubit example, the authors show the counterintuitive result that placing the qubit in the $|0\rangle$ state during most of the experiment and applying the $u_x$ and $u_y$ fields near the end of the estimation period results in a better estimate of the eigenfrequency of the system than placing the system in the $|+\rangle$ state during the entire estimation period as is commonly done in conventional Ramsey type estimation of the frequency.
One assumption the authors seem to make implicitly is that the controls $u_x$, $u_y$ and $u_z$ are ideal. In practice, the quantity that is being estimated (the frequency) is exactly the same quantity that is used to determine the phases $x$ and $y$ of the control fields $u_x$ and $u_y$. As such, I would expect the method to work only because the phase tracking that is normally done by the qubit state in the $|+\rangle$ state to now be done by the "ideal" control fields.
If the authors could explain intuitively how this is not the case, and the error in the control would not affect the estimate it would improve the manuscript significantly.
Although the 2-qubit example is not directly about estimating the qubit frequencies but rather the coupling, it is not clear that this estimate is not also affected by frequency errors in the control fields.
Some minor points.
When reading the abstract and introduction it was not immediately clear that this manuscript is about using optimal control to determine what fields to apply to acquire better estimates rather than using optimal control to determine what driving fields would optimally implement a certain operation. This is mostly because finding optimal control pulses is a very common application these days. Being a bit more explicit could help the casual reader identify what this paper is about quicker.
The images of the driving fields could be improved by adding a few images of the Bloch sphere depicting the (ideal) state of the qubit during different parts of the protocol. This would also help the intuitive understanding.
In the examples it is not clear to me if there is a discrete control pulse near the end of the estimation period $T$ or if the fields are brought to oscillate during the entire estimation period. The textual description seems to suggest the former while the figures (typically figure b) seem to suggest the latter.

---

## Round 1 · Referee Report · Anonymous (Referee 3) · 2022-6-13

Strengths
interesting results
Report
The authors proposed to apply quantum control for the improvement of the precision of parameter estimation. They demonstrated that by applying the optimal quantum control, the quantum Fisher information, which quantifies the local precision limit for the estimation, can be significantly improved. The method also shows other advantages, such as the time stability in maintaining the maximum QFI over long time spans. The results have wide applications in quantum sensing and the determination of the models for quantum systems. I thus recommend the publication.
In the current study, the authors assume perfect controls. Although controls with systematic errors can be dealt with by many existing techniques, such as robust controls and composite pulses, the authors should at least mention it and have a discussion on this issue, which is important for practical applications.
In the current study, the authors assume perfect controls. Although controls with systematic errors can be dealt with by many existing techniques, such as robust controls and composite pulses, the authors should at least mention it and have a discussion on this issue, which is important for practical applications.

---

## Round 2 · Referee Report · Anonymous (Referee 3) · 2022-9-4

Report

The authors have addressed all the concerns in the revised manuscript. I recommend the publication.

---

## Round 2 · Referee Report · Anonymous (Referee 2) · 2022-9-8

Report

The authors have addressed the concerns raised in the original review.

---

## Round 2 · Author Response

Dear Editor,

We have revised the manuscript "Optimal control for Hamiltonian parameter estimation in non-commuting and bipartite quantum dynamics" in the light of the reviewers' comments. We believe we have been able to respond to all such comments and to produce an improved version of the paper, which we hereby resubmit.

Let us list the changes in this new version below, along with the comments that prompted them, when relevant:

• All reports point out that perfect controls are assumed and that this should be discussed or at least mentioned. In our study, robustness has been dealt with through the (d) parts of our plots, where the performance of controls optimised for wrong estimates of the parameter is depicted, yielding tolerances between 5 and 20% of the parameter, depending on specific cases. In the revised version of the manuscript, we have linked the application of control optimised for wrong parameters explicitly to the robustness of the scheme by carrying out further analysis and obtaining direct information on the tolerance in terms of amplitude and timing mismatches. Hence, we have much expanded the discussion of the robustness plots with dedicated paragraphs at the end of sections 3.1 and within section 4.1, and with a comprehensive final discussion in the conclusions on page 16.

• Reviewer 2 asks whether, in single-qubit frequency estimation, the advantage of having the qubits dwell around the |0> rather than |+> state might be an artefact due to assuming perfect control fields. This is not the case, as discussed in our revised discussion of robustness already reported above. However, we have endeavoured to better explain our results by adding a detailed descriptions of the controlled dynamics and of their advantages, for both dephasing and relaxation: "In this case, as discussed in [42], the optimal controls exploit the decoherence free subspace of the states diagonal in the Pauli-z eigenbasis (i.e., the z-axis of the Bloch sphere)." and "Under relaxation, the states are first steered towards the (decoherence free) ground state and then, after a few precessions around the z-axis, towards the x-y plane. Although the protection from decoherence achieved through this trajectory is not as complete as in the case of dephasing, where a decoherence free subspace is used, states with different parameters separate faster than in the dephasing free subspace, resulting in a better distinguishability and thus a higher maximum QFI value (the reader should bear in mind that the QFI reflects the cumulative effect of dynamics upon the evolving state, rather than the information in the initial state, whose preservation is immaterial here)."

• As both reviewer 1 and 2 requested, the arguments above are now supported by including a visualisation of the controlled Bloch vectors, which we have included also for the non-commuting case and 2-qubit cases by introducing the new Figs. 2, 5 and 12. This should greatly boost the accessibility and clarity of our study.

• Reviewer 1 finds the better performance achievable under relaxation (over dephasing) surprising. This aspect has been addressed in detail through the additional explanations reported at the previous point.

• The maximal QFI as a function of decoherence rates, requested in report 1, was already addressed in Basilewitsch et al.'s paper (Ref.[42]), to which the reader is now explicitly referred at the beginning of page 6.

• Reviewer 2 points out that it is not clear at first sight whether the controls are optimised for estimation or to achieve controlled operations: the addition of "in the estimation of Hamiltonian parameters" in the abstract should dispel this confusion.

• The introductory section has been expanded, and 10 additional references have been included.

• Reviewer 2 is asking whether a final discrete pulse is applied: it is not, as the reviewer correctly surmises. We believe this confusion is down to the old description in the relaxation section, which was confusing. We have adapted the part to not mention pi/2 pulses, which should clarify the issue.

• Reviewer 1 asks to comment as to the respective advantage of control vs mere repetition. To this aim, we have added the following sentence in the conclusions: "Notice also that, in extracting information about the parameter, the advantage of controlled dynamics over repeated uncontrolled ones would ultimately depend on the latter's cost in specific circumstances (although of course controls and repetition can be combined, attaining better information rates)."

Sincerely,
Shushen Qin, Marcus Cramer, Christiane P. Koch and Alessio Serafini

---

## Round 2 · List of Changes

1. Stated the aim of the paper more explicitly in the abstract.
  2. Included a paragraph on the preference of a sequential estimation scheme over a parallel scheme and 10 additional references in the introduction.
  3. Expanded on the description of the controlled dynamics in section 3.1 and 3.2.
  4. Included discussion on robustness of control fields at the end of section 3.1 (bottom of the left column on page 6), within section 4.1 (center of the right column on page 12) and in the conclusion.
  5. Rephrased the description of control fields in section 3.2 to remove mention of pi/2 pulses.
  6. Emphasised that the performance of the controlled estimation is independent of the choice of probe states in the conclusion.
  7. Included Bloch vector plots (Fig. 2, 5 and 12).

---

## Editorial Decision

published